# Lumbar Intervertebral Disc Herniation: Annular Closure Devices and Key Design Requirements

**DOI:** 10.3390/bioengineering9020047

**Published:** 2022-01-19

**Authors:** Alexandra Alcántara Guardado, Alexander Baker, Andrew Weightman, Judith A. Hoyland, Glen Cooper

**Affiliations:** 1School of Engineering, University of Manchester, Manchester M13 9PL, UK; alexandra.alcantaraguardado@manchester.ac.uk (A.A.G.); andrew.weightman@manchester.ac.uk (A.W.); 2Royal Preston Hospital, Preston PR2 9HT, UK; alexbaker@doctors.org.uk; 3School of Biological Sciences, University of Manchester, Manchester M13 9PL, UK; judith.a.hoyland@manchester.ac.uk

**Keywords:** intervertebral disc, lumbar IVD herniation, annular closure device, design specification

## Abstract

Lumbar disc herniation is one of the most common degenerative spinal conditions resulting in lower back pain and sciatica. Surgical treatment options include microdiscectomy, lumbar fusion, total disc replacement, and other minimally invasive approaches. At present, microdiscectomy procedures are the most used technique; however, the annulus fibrosus is left with a defect that without treatment may contribute to high reherniation rates and changes in the biomechanics of the lumbar spine. This paper aims to review current commercially available products that mechanically close the annulus including the AnchorKnot^®^ suture-passing device and the Barricaid^®^ annular closure device. Previous studies and reviews have focused mainly on a biomimetic biomaterials approach and have described some mechanical and biological requirements for an active annular repair/regeneration strategy but are still far away from clinical implementation. Therefore, in this paper we aim to create a design specification for a mechanical annular closure strategy by identifying the most important mechanical and biological design parameters, including consideration of material selection, preclinical testing requirements, and requirements for clinical implementation.

## 1. Introduction

### 1.1. Background

Low back pain is among the leading causes of disability in the world [1]. Injury or degeneration of intervertebral discs (IVDs) in the spine can be a cause of this disability. The IVD has a complex structure to enable the biomechanics of human movement. The IVD consists of a central nucleus pulposus, an irregular meshwork of collagen II fibrils with an osmotic swelling pressure created by large amounts of proteoglycan aggrecan aggregated along chains of hyaluronan. This is surrounded by the annulus fibrosis, collagen I fibers oriented in oblique angles in a heterogeneous structure in distinct lamellae, with the collagen fibers laid up in alternating oblique angles in each consecutive lamella to form an angle-ply architecture [2,3,4,5,6]. The IVD structure is illustrated in Figure 1.

Degenerative changes in the intervertebral discs (IVDs) can cause a loss of hydration in the nucleus pulposus reducing IVD height. This leads to a greater load transfer to the surrounding annulus fibrosus (AF) creating microstructural damage in the fibers that with time can develop into bigger tears [7,8]. Defects in the AF compromise the overall stability of the IVD and change the normal position of the nucleus causing bulging and, in more severe cases, IVD herniation. Lumbar IVD herniation is one of the most common degenerative spinal diseases causing low back pain and sciatica [9,10]. IVD herniation is defined as the localized or focal displacement of IVD material beyond the limits of the IVD space [11]. The condition affects approximately 1–3% of the general population [12], with 15–20% of these cases requiring surgical intervention [13]. These can include microdiscectomy, lumbar fusion, prosthetic IVD replacement (total disc replacement), and several other minimally invasive approaches [14].

Spinal fusion aims to reduce pain by restricting the motion of the affected vertebral segment(s), but postoperative biomechanical alterations can result in loss of normal range of motion, reduced shock absorption, instability, and spinal misalignment [15,16,17]. These changes in turn cause abnormal stress/strain and increased intradiscal pressure in the adjacent segments [16,18]. Ultimately this can lead to degeneration in IVDs adjacent to the fused segment(s). Total IVD replacement or IVD arthroplasty is a potential alternative to spinal fusion. It aims to restore IVD height and normal movement to the spine and prevent early degeneration of adjacent segments. However, in reality these artificial IVDs have limited mobility compared to normal biological IVDs [19] and may cause further complications by dislodging from the vertebral bodies or releasing debris from the wear and friction of the implant. Open discectomy and microdiscectomy remove only the protruding IVD material [20,21]; these methods have high success rates ranging between 75% and 95% [21]. However, the amount of nuclear material removed and ligament damage play a significant role in intervertebral translational instability [22]. 

### 1.2. The Importance of Annular Closure 

Untreated defects in the AF contribute to higher reherniation rates [23,24] and changes in the stability of the adjacent segments [25]. Recently, Miller et al. demonstrated that patients with large annular defects (≥6 mm) after lumbar discectomy have higher risk for symptom recurrence and reoperation compared to those with small annular defects [24]. This is because IVDs have a very limited capacity for self-healing upon degeneration or injury [26]. If any annular healing occurs after discectomy, it is believed to happen slowly and ultimately results in biomechanically inferior tissue with reduced capacity to accommodate tensile forces [19]. This in turn may trigger reherniation under lower biomechanical stresses in patients with large postsurgical annular defects, which comprise approximately 30% of those treated with lumbar discectomy [26,27]. Current research in tissue engineering is investigating biomimetic approaches to replicate the lamellar structure and induce regeneration [28]; however, the avascular nature of this region and the low technology readiness level of existing research strategies mean that these approaches are likely to be decades away from clinical implementation. A recent review on advanced regeneration strategies of the annulus fibrosis concluded that although injectable hydrogels may be a possible solution there are problems integrating them with native tissues and that key challenges are to understand the IVD herniation pathway with a particular focus on structural aspects of the annular fibrosis lamella and interlamellar matrix [29]. On the other hand, repair has been implemented in the clinical environment using mechanical closure devices [23,30,31]. Some of these have attempted to mechanically close annular defects using different sutures [25,31,32,33,34,35,36,37], mesh-like devices [38], cyanoacrylate or other adhesives [25,39,40,41,42,43,44,45,46], and patch- and plug-like implants [47,48,49] with mixed results. None of these have been tested long-term and most have only been tested in vitro. In addition to these closure strategies, several commercial products have been CE marked and FDA approved to mechanically close the annulus. These include the Inclose Surgical Mesh System and the Xclose Tissue Repair System (Anulex Technologies, Inc. Minnetonka, MN, USA), the AnchorKnot^®^ suture-passing device (Anchor Orthopedics XT Inc., Mississauga, ON, USA), and the Barricaid^®^ Annular Closure Device (Intrinsic Therapeutics, Woburn, MA, USA).

### 1.3. Commercial Annular Closure Devices

#### 1.3.1. Inclose and Xclose Systems

The Inclose Surgical Mesh System and the Xclose Tissue Repair System (Anulex Technologies, Inc. Minnetonka, MN, USA) were both commercially available and marketed as annular closure devices (Figure 2, top); both devices received US Food and Drug Administration (FDA) 510(k) clearance in 2005 and 2006, respectively [36,38]. The Inclose system consisted of an expandable braided mesh-like patch and two polyethylene terephthalate (PET) suture tethers (Anchor Bands by Anulex Technologies Inc., Minnetonka, MN, USA). Evidence of the safety and use of Inclose is limited; no peer-reviewed literature is available at the time of publication of this article. On the other hand, the Xclose Tissue Repair System was more widely used as an annular closure device. However, this device was not demonstrated to significantly reduce reherniation rates [50]. Both systems were withdrawn from the market after multiple complications were reported [33]. 

#### 1.3.2. AnchorKnot^®^ Tissue Approximation Kit

The AnchorKnot^®^ suture-passing device (Anchor Orthopedics XT Inc., Mississauga, ON, USA) is designed to close the annular defect after partial or total nucleotomy (Figure 2, middle). It consists of a Kerrison-shaped device that delivers a 2-0 ultra-high molecular-weight polyethylene (UHMWPE) suture with a Dines knot to the AF [31]. Bateman et al. showed no tissue reaction to the suture material and no NP extrusion at any of the sutured levels when used in an in vivo porcine model. However, magnetic resonance imaging (MRI) volumetric assessment showed reduced volume in both unrepaired and sutured IVDs compared with unoperated control IVDs, but this study was not statistically significant [31].

#### 1.3.3. Barricaid^®^ Annular Closure Device

The Barricaid^®^ Annular Closure Device (Intrinsic Therapeutics, Woburn, MA, USA) is another commercially available implant used in lumbar discectomies (Figure 2, bottom). The device consists of a woven polyethylene terephthalate (PET) mesh attached to a titanium bone anchor fixed in position into either of the adjacent vertebral bodies of the IVD. The flexible polymer mesh forms a mechanical barrier that closes the annular defect and prevents subsequent migration of the nuclear material. This device is available in 8, 10, and 12 mm widths and comes preloaded onto a disposable insertion tool that has already been sterilized [51]. Clinical outcomes of the Barricaid^®^ device show promising results [52], demonstrating that this technique could address some IVD height issues. It prevents same level IVD herniation recurrence and decreases scores in the visual analogue scale (VAS) and Oswestry disability index (ODI) [53]. However, the complex loading modes the Barricaid^®^ has to withstand have caused the implant to occasionally loosen [54,55], and in one reported case it came into contact with the spinal nerves causing significant pain in the patient [56]. An additional challenge with this type of nondegradable implant is that long-term consequences (<5 years) of implantation are still unknown [57]. Nonetheless, the Barricaid^®^ implant is thus far the only commercial device to have some clinical evidence of success at addressing the challenging problem of annular closure.

## 2. Design Requirements for an Annular Closure Device

### 2.1. Previous Efforts at Developing an Annular Repair/Regeneration Strategy

Researchers have mostly studied the use of biodegradable or bioresorbable implants to close annular defects while aiding with tissue regeneration over a finite period. However, the extremely low cellularity of adult cartilage constitutes a serious problem when injured [26,59]. In several animal models with surgically injured IVDs, healing has been demonstrated to occur in the outer annulus but not towards the inner part of the IVD [37,60]. This is believed to be due to the outer annulus having four times the cell density of the nucleus in adult human IVDs [61] implying that the tissue may have some ability to heal, though this has not been conclusively proven [60].

Evidence suggests that annular healing proceeds in an outside to inside direction [37]. As discussed above, if any annular healing occurs at all it is hypothesized to happen slowly with tissue that has a reduced capacity to accommodate everyday biomechanical forces [24]. Should a repair/regeneration approach be selected, then degradation time of the implant should coincide with the healing process to ensure proper remodeling of the tissue [59]. These parameters are still unknown for both human and animal IVD tissue. Additionally, the mechanical properties of the implant should be adequate to promote regeneration during the patient’s everyday activities, and any change in mechanical properties due to degradation should preserve compatibility with the healing or regeneration process [59]. It is estimated that the spine undergoes approximately 100 million flexion cycles during a lifetime [62]. In the case of spinal implants, 10 million (1 × 10^7^) cycles is considered to be the minimal life length, but 30 million cycles is considered optimal [63]. Any biodegradable/bioresorbable strategy should be able to ensure that regenerated tissue would be able to withstand the same number of cycles.

Current biodegradable/bioresorbable strategies to repair the adult human IVD are highly likely to fail before sufficient tissue regeneration occurs. From a biomechanical point of view IVD regeneration may only occur in if the following conditions are met: restoration of normal physiological range of motion, normal lordosis, spinal balance achieved, IVD height restored, normal intradiscal pressure, and normal IVD load distribution [63]. None of these are achieved by current implants, biological repair strategies, or tissue engineered approaches [64] suggesting that a permanent (non-resorbable) mechanical closure device may be the most suitable option for this application. 

### 2.2. Aims and Scope

There have been many excellent reviews describing the necessary biological parameters for annular repair/regeneration [27,65,66,67,68]; suitable biomaterials [69]; tissue engineered scaffolds [64]; and the challenges and strategies [57,70] for implementation. Long et al. made an important breakthrough by identifying, quantifying, and proposing a testing paradigm for hydrogel-based strategies for AF repair [71] as shown in Table 1. However, in this paper we focus on mechanical annular closure devices as they are closer to clinical implementation. To do this, we (i) identify the key design requirements; (ii) highlight material considerations; (iii) discuss preclinical testing requirements; and (iv) review challenges for clinical implementation. 

### 2.3. Key Requirements for an Annular Repair Strategy

#### 2.3.1. Mechanical Requirements

The AF is composed of concentric lamellae of collagen fibers embedded into a proteoglycan matrix or ground substance. In the human AF, there are approximately 15–20 lamellae, each with around 40 collagen bundles [72] alternating from 45° to 25° with an average value of 28° [73]. The highly organized lamellar structure allows it to distribute and absorb large spinal load that occurs in complex combinations of tension, compression, torsion, shear, and bending. Intradiscal pressure tensions the annular fibers and supports the endplates. It is the main contributing factor to adequate IVD height and tissue stiffness during axial compression [74]. Any reduction in the pressure reduces IVD volume through IVD height reduction; this changes the stress distribution in the IVD causing increases in stress concentrations within the IVD and leads to increased shear forces in the nucleus [74]. Therefore, AF repair devices must withstand intradiscal pressure when prone, when supine immediately after surgery, and sitting and standing pressures shortly after surgery [71]. Additionally, for an individual to continue their activities of daily living (e.g., walking, standing, climbing stairs), the range of motion must be restored in all six degrees of freedom (flexion–extension, lateral bending, and axial rotation) within an acceptable range. Reference values for range of motion, intradiscal pressure, and IVD height for healthy spines are shown in Table 2.

#### 2.3.2. Biological Requirements

In the IVD’s central regions, the nucleus and inner AF are supplied by capillaries that arise in the vertebral bodies, penetrate the subchondral bone, and terminate at the endplates [76,77]. In the outer region, cells require the blood supply in the outer AF of the IVD to receive their nutrients and metabolites. Small molecules such as glucose and oxygen then reach the cells by diffusion under gradients established by the balance between the rate of transport through the tissue to the cells and the rate of cellular demand [76,77]. Therefore, it is crucial that an annular repair strategy minimizes disruption of the blood supply in the outer annulus and does not cause significant damage or lesions to the endplates that will compromise nutrient supply [78]. Furthermore, preservation of the endplates is crucial, particularly the caudal (bottom) endplate. Caudal endplates have lower bone mineral density, are thinner than the cranial endplates, and have significantly more openings, which could contribute to the small thickness of this endplate and its susceptibility to fractures [79]. In the case of the Barricaid^®^ device, increased prevalence of new endplate lesions and loss of surrounding bone were observed with use of the implant compared to the controls; endplate lesions close to the flexible polymer component were considerably much larger [55]. This has been suggested to be a result of the PET mesh being in contact with the adjacent vertebral body as a result of IVD height loss [80]. These lesions appeared to stabilize over time, and no vertebral fractures occurred within the five year period [55]. However, it may be sensible to recommend that surgeons place the titanium anchor in the lower vertebrae to avoid causing substantial long-term damage to the caudal endplate. Furthermore, implants that consider using a similar fixation strategy to the Barricaid^®^ device (using the upper or lower vertebrae to secure the implant in place) may need to carefully consider the place of attachment and its biological consequences. 

Additionally, the aging process and/or IVD degeneration reduces the diffusion of nutrients and metabolites causing the accumulation of lactic acid in the center of the IVD [81]. This in turns lowers pH from a healthy ~7.1 to values of 6.5–5.7 [81]. Low pH has been reported to reduce cell viability and proteoglycan and collagen synthesis in the IVD [81]. This would suggest that maintenance of a normal physiological pH would be beneficial to prevent further IVD degeneration and herniation.

#### 2.3.3. Material Requirements

Any biomaterial used must be biocompatible and non-cytotoxic. Additionally, any material should ideally possess mechanical properties similar to those of the surrounding annular tissue to encourage natural load distribution throughout the intervertebral and adjacent spinal segments. If the material modulus is too high, it will not deform accordingly, and the majority of the load would be concentrated on the implant; this increases the probability of the implant fracturing and weakening the surrounding tissue [46]. On the other hand, if the material modulus is too low, the implant would not be able to support the load, putting more strain onto the adjacent tissue and establishing the possibility of fragments (or the whole implant) being expelled into the spinal canal and damaging the nerves [71,82].

### 2.4. Preclinical Testing

In vitro testing provides a controlled method for investigating biocompatibility and preliminary mechanical tests of a scaffold or promising annular repair materials. Both mechanical and biological in vitro testing have several challenges to overcome. IVD sizes vary across species and according to location within the spine. Most commonly used in vivo animal models range from small rodents to rats, rabbits, dogs, goats, sheep, primates [83], and more recently kangaroos [58]. One of the main issues when using animal models is that the majority of them are quadrupedal, and the few bipedal models available (primates and kangaroos) present ethical dilemmas that prevent their use in most research institutions [83]. Biomechanics are also significantly different; hence, it is hypothesized that the loads exerted on the lumbar IVDs of large animals by the surrounding structures (muscles and ligaments) may be even greater than those in human IVDs resulting from the bipedal stance due to the increased difficulty of stabilizing a horizontally aligned spine versus a vertically balanced spine [83]. However, use of animal tissues can help in understanding how aspects of testing techniques influence the results of experiments on human tissue [84]. 

Computational and finite element (FE) modeling have become powerful tools to evaluate performance of current and novel medical devices. The initial and long-term performance of a device could be predicted with anatomically accurate human models which can influence device design and optimization reducing the need for in vivo animal testing. However, one of the main challenges with current models is that failure mechanisms of the IVDs are quite a complex task to model due to the tissue’s complex structure. Few studies have focused on modeling annulus failure mechanics of human tissue. Early models include that of Goel et al. to evaluate delamination in the lamellae based on interlaminar shear stresses [85] and evaluation of damage in a hyperelastic anisotropic model of the annulus by Eberlein et al. [86]. Quasim et al. studied damage initiation and progression in the AF region under various cyclic loading conditions using a 3D poroelastic model [87]. However, with this model only damage to the tissue will occur in the direction of tensile principal stress. Furthermore, they did not consider the changes in the viscoelastic characteristics of the annulus [87]. More recently, Shahraki et al. used the Tsai–Wu criterion and the maximum normal stress to predict damage initiation and propagation of the AF tissue under different loading conditions [88]. Nonetheless, this model does not account for water content and porosity of the IVD. Furthermore, the annulus region was considered as a homogeneous material in terms of fiber orientation and density, but fiber orientation in the tissue changes from the inner to the outer part of AF. Additionally, viscoelasticity was not considered, and strength of the material was considered under static loading only [88].

To the authors’ knowledge there are no finite-element analyses of tissue engineered scaffolds for AF or IVD repair to date; however, Moroni et al. fabricated 3D meniscal scaffolds with porosity and architecture that mimic native tissue mechanical properties [89]. The anatomical 3D scaffolds were experimentally analyzed to tailor their mechanical properties to match those of natural menisci and numerically investigated with FE analysis to determine whether they were mechanically robust enough to be used for meniscus reconstruction [89]. While menisci and IVDs are quite different, they are both anisotropic, viscoelastic tissues, thus proving that it would be beneficial to use FE analysis to test and optimize a scaffold’s design. It could also help in reducing time and costs if it is used to test a preliminary design to determine optimal structural, mechanical, and physical properties to direct specific cell function [90].

### 2.5. Considerations for Clinical Translation

#### 2.5.1. Sterilization

For any medical device, proper sterilization is crucial [60]; thus, the annular repair strategy must be sterilizable. Depending on the material used and type of repair strategy used this may prove very challenging. For example, in the case of hydrogel-based strategies sterilization must be performed carefully because these materials may be sensitive to the sterilizing agents such as heat and radiation [91]. Water content in the hydrogel can facilitate chemical bond breakdown which can result in changes of material properties, degradation and/or decomposition of the material, discoloration, embrittlement, odor generation, and promote further crosslinking or induce toxic effects [91]. 

#### 2.5.2. Delivery and Attachment

As discussed in Section 1.2, current minimally invasive procedures, which are the best surgical option to treat lumbar IVD herniation, limit the amount of soft tissue dissection and minimize excessive scarring near the nerve root [86]. Thus, an AF repair device implanted during a microdiscectomy should be minimally invasive and create minimal soft tissue damage during the procedure. In this case, injectable scaffolds could allow easy filling of irregularly shaped defects [65]. One of the greatest challenges in cartilage tissue engineering is achieving fixation or functional integration into the native tissue [92]. The optimal device must be safely attached to the tissue and must remain in place. Adhesive patches based on bioinspired architectures such as the gecko-/beetle-inspired mushroom-shaped architectures, endoparasite-like microneedles, octopus-inspired suction cups, and slug-like adhesive [58] may be the answer to achieve adequate fixation to the native AF, should this approach be adopted.

#### 2.5.3. Postoperative Imaging

Magnetic resonance imaging (MRI) is a useful imaging technique capable of producing high quality images of the human body and individual tissues. It allows non-invasive assessment of IVD degeneration/herniation and provides excellent spatial resolution and tissue characterization without exposing patients to the potential risks of ionizing radiation and iodinated contrast agents [93]. When monitoring endplate lesions pre- and postoperatively, MRI has proved equally useful [91]. The American Society for Testing and Materials (ASTM) categorizes implants into [94]:“MRI Safe—an item that poses no known hazards resulting from exposure to any MR environment. MR Safe items are composed of materials that are electrically nonconductive, non-metallic, and nonmagnetic.MR Conditional—an item with demonstrated safety in the MR environment within defined conditions.MR Unsafe—an item which poses unacceptable risks to the patient, medical staff, or other persons within the MR environment.”

Safe imaging of the operated area during follow-up appointments is vital in case of a patient experiencing adverse effect or symptom recurrence. Therefore, according to ASTM standards an annular repair strategy must be MRI safe or MRI conditional. For example, even though the Barricaid^®^ device (Intrinsic Therapeutics, Woburn, MA, USA) has metallic components, it can be safely tested with static magnetic fields of 1.5 and 3 Tesla (maximum spatial gradient magnetic field of 3000 Gauss/cm or less) [51]. 

#### 2.5.4. Same Level Symptomatic Reherniation

Symptomatic IVD reherniation is the most common cause of reoperation after primary IVD surgery [95]. Although there are many theories as to what increases a patient’s chance for reherniation, no one factor has been identified consistently in the literature [96]. Reherniation rates of surgically treated patients with IVD herniation vary greatly across the literature, from 3% to 18% [97] or between 0.5% and 25% [98]. It is therefore difficult to specify an acceptable range of percentage risk; however, recent clinical outcomes of the Barricaid^®^ device vs. controls show that the cumulative incidence of symptomatic reherniation was 8.4% vs. 17.4% at 1 year, 10.7% vs. 23.4% at 2 years, and 14.8% vs. 29.5% at 3 years [99]. A negligible or low reherniation risk would be optimal. 

#### 2.5.5. Device Loosening, Failure, and Safe Removal

Clear instructions must be given to the surgical team regarding implant delivery to ensure correct placement and that patients are given the necessary long-term postoperative care guidelines to ensure the AF repair device’s appropriate function. However, in case a patient begins to present adverse effects or pain related to implantation of the device, it is necessary to have a procedure for the safe removal of the device.

#### 2.5.6. Pain Score Improvement

There are currently at least 28 scoring systems available for the evaluation of low back pain [52]. In the lumbar spine, the visual analogue scale (VAS) and the Oswestry disability index (ODI) are used most often. The VAS consists of a straight line with the endpoints defining extreme limits such as ”no pain at all” (score of 0) and ”pain as bad as it could be” (score of 10; 100 mm scale) [100]. In the ODI, the total score ranges from 0% to 100%, with 0% representing no disability and 100% representing maximum disability. For example, a total score between 0% and 20% means minimal disability; between 20% and 40%, moderate disability; between 40% and 60%, severe disability; between 60% and 80%, crippled; and between 80% and 100%, bedbound or symptom magnification [52]. For VAS and ODI scores, a decrease of at least 20mm and 15 points, respectively, is considered a success [55]. In the case of current commercial annular closure devices there was no significant difference in follow-up ODI and VAS scores for both back and leg at 90 days and 2 years when an annular closure device was used compared to microdiscectomy only [52].

### 2.6. Current Testing Standards

At present, there are no published standards that specify the requirements for annular closure devices/implants. However, there are related standards which cover mechanical and biological requirements for total IVD prostheses, nucleus replacements, material evaluation and characterization, and imaging requirements for medical devices. Table 3 shows excerpts from standards which are most relevant to the present application [101,102,103,104,105,106,107,108,109,110,111,112,113,114]. For example, BS ISO 18192 series define mechanical in vitro testing procedures to simulate and evaluate lumbar spinal IVD prostheses wear under adverse conditions [101,102,103]. These also specify the minimum implant life to be at least 10 million (10 × 10^6^) cycles or 10 years. BS EN ISO 10993 Parts 1–20 are comprehensive guidelines to evaluate medical devices ranging from risk management process to animal testing [109]. For material characterization ASTM F981-04(2016) provides a series of experimental protocols for biological assays of tissue reaction to nonabsorbable biomaterials for surgical implants where the materials will reside in bone or soft tissue for over 30 days [110]. This table is not exhaustive; there exist many current BS ISO and ASTM standards describing the guides and standard practices for the characterization and testing of medical devices, biomaterials, and tissue engineered medical products (TEMPs). The authors recommend that researchers conduct a preliminary search in the respective database before planning any tests.

## 3. Conclusions and Recommendations

Following microdiscectomy, AF repair has the potential to improve medium- and long-term outcomes. Several previous studies have highlighted the potential of annular repair in order to prevent future reherniation. However, available commercial options for surgical repair of the AF are limited. The AnchorKnot^®^ suture-passing device (Anchor Orthopedics XT Inc., Mississauga, ON, USA) and the Barricaid^®^ Annular Closure Device (Intrinsic Therapeutics, Woburn, MA, USA) are the only CE-marked and FDA approved devices currently in the market, with the latter having demonstrated clinical success for a reported period of up to 5 years. Current research has focused greatly on the use repair/regeneration therapies, but due to the limited healing properties of the IVD and the clinical readiness of research solutions, these approaches are not likely to be clinically implemented soon. Furthermore, most strategies do not follow a clear design process and solely concentrate on exploring novel permutations and combinations of materials without these being rigorously mechanically tested. It is evident that that for a mechanical AF closure strategy to be successful, there is need for a concise design specification. This review outlines some of the key requirements from the literature which could create a design specification including mechanical, biological, material, preclinical, and clinical considerations for a successful AF closure strategy in Appendix A. These key requirements may help make informed decisions for future research and design concept evaluation for an effective AF closure device. 

## Figures and Tables

**Figure 1 bioengineering-09-00047-f001:**
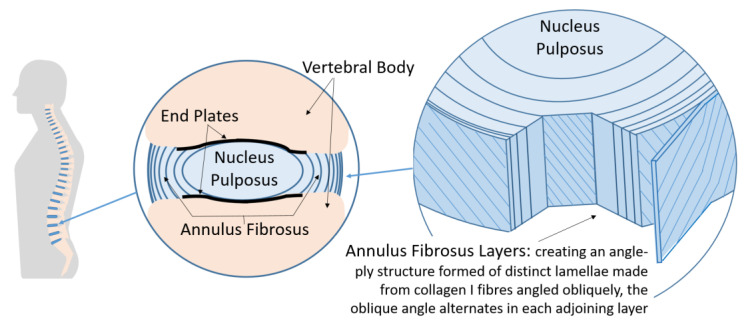
Intervertebral disc structure (IVD). (**Left**) The spine within the body containing the vertebrae (bone) separated by IVDs. (**Middle**) Sectional view of the IVD. (**Right**) Cutaway view of the IVD showing the angle-ply structure of the annulus fibrosus, adapted from [6].

**Figure 2 bioengineering-09-00047-f002:**
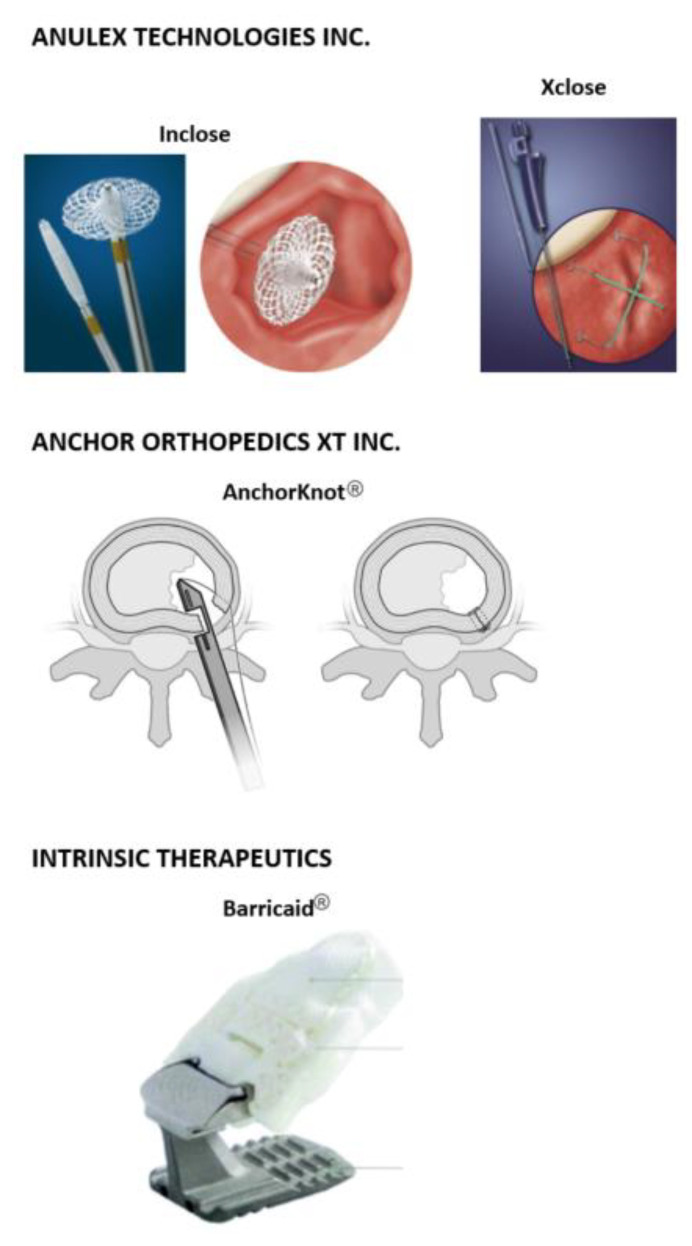
Commercial mechanical annular closure devices. (**Top**) Inclose Surgical Mesh System and the Xclose Tissue Repair System (Anulex Technologies, Inc. Minnetonka, MN, USA) [27]. (**Middle**) The AnchorKnot^®^ suture-passing device (Anchor Orthopedics XT Inc., Mississauga, ON, USA [31]. (**Bottom**) The Barricaid^®^ Annular Closure Device (Intrinsic Therapeutics, Woburn, MA, USA) [58].

**Table 1 bioengineering-09-00047-t001:** Suggested design criteria for a hydrogel-based strategy and recommended values from Long et al. [71].

Design Criteria for Hydrogel-Based Repair Strategies	Recommended Design Parameters	
	Parameter	Recommended Value
Device adhesion testing	IVD pressure, after implantation	1.5 MPa
Similar biomaterial properties to native tissue	IVD pressure, maximal	2.3 MPa
	Tensile modulus, axial	0.5–1 MPa
Biocompatibility and cytotoxicity	Compressive/tensile strain	28%/65%
Biomaterial degradation rate	Axial stiffness of restored IVD	1.5–2 kN/mm
Biomechanics evaluation	Torsional stiffness of restored IVD	3.2 Nm/deg
Reherniation risk	Tensile modulus, circumferential	11–29 MPa
	Aggregate modulus	0.4–6 MPa
	Shear modulus	0.1–0.28 MPa

**Table 2 bioengineering-09-00047-t002:** Reference values for ROM, IDP, and IVD height in normal asymptomatic patients, data taken from [71,75].

Range of Motion				
	Mean angle (degrees)			
Flexion	6°–13°			
Extension	1°–5°			
Lateral bending	2.9°–11°			
Axial rotation	2°–3°			
**Intradiscal Pressure**				
	Mean (MPa)	Maximum (MPa)		
Prone	0.22	0.41		
Sitting	0.75	1.50		
Standing	0.59	1.07		
**IVD Height (mm)**				
	Anterior		Posterior	
	Male	Female	Male	Female
L1-L2	7.48 ± 1.5	5.92 ± 1.2	4.91 ± 1.2	4.34 ± 0.9
L2-L3	8.54 ± 1.5	7.15 ± 1.5	5.65 ± 1.4	5.11 ± 1
L3-L4	9.58 ± 1.7	8.08 ± 1.4	6.02 ± 1.4	5.57 ± 1.2
L4-L5	10.89 ± 2.1	9.76 ± 1.9	6.11 ± 1.3	5.97 ± 1.4
L5-S1	11.8 ± 2.6	11.22 ± 2.5	5.24 ± 1.4	5.2 ± 1.4

**Table 3 bioengineering-09-00047-t003:** Relevant standards and recommendations for testing.

Category	Standard	Scope
Mechanical	BS ISO 18192-2:2010: Implants for surgery. Wear of total intervertebral spinal disc prostheses. Nucleus replacements [101]	Defines an in vitro test procedure to simulate and evaluate lumbar spinal IVD prostheses wear under adverse impingement (point at which two opposing components collide or restrict motion) conditions for nucleus replacements.
	BS ISO 18192-3:2017: Implants for surgery. Wear of total intervertebral spinal disc prostheses. Impingement-wear testing and corresponding environmental conditions for test of lumbar prostheses under adverse kinematic conditions [102]	Defines an in vitro test procedure to simulate and evaluate lumbar spinal IVD prostheses wear under adverse impingement (point at which two opposing components collide or restrict motion) conditions. Minimum axial load of 300 N (or equivalent to generate a 7.5 Nm moment). Cycle limit of 10 × 10^6^ cycles.
	BS ISO 18192-1:2011+A1:2018: Implants for surgery. Wear of total intervertebral spinal disc prostheses. Loading and displacement parameters for wear testing and corresponding environmental conditions for test [103]	Defines a procedure for the relative angular movement between articulating components and specifies a pattern of the applied force, speed and duration of testing, sample configuration, and test environment for use for the wear testing of total intervertebral spinal sic prostheses. Fluid test medium of calf serum at a concentration of 20 g ± 2 g protein/L. Minimum of 6 test specimens. For lumbar IVD prostheses, the testing machine must produce angular displacements of (min–max): −3° to 6° for flexion/extension and 2° to −2° for axial rotation and lateral bending. Load parameters of (min–max): 600–2000 N. Cycle limit of 10 × 10^6^ cycles.
Mechanical (continued)	ASTM F1717-18: Standard Test Methods For Spinal Implant Constructs In A Vertebrectomy Model [104]	Covers the materials and methods for the static and fatigue testing of spinal implant assemblies in a vertebrectomy model. Three static mechanical tests and one dynamic test evaluate the spinal implant assemblies. The three static mechanical tests are compression bending, tensile bending, and torsion. The dynamic test is a compression bending fatigue test.
	ASTM F2789-10(2015): Standard Guide for Mechanical and Functional Characterization of Nucleus Devices [105]	Describes various forms of nucleus replacement and nucleus augmentation devices. The tests for characterizing the performance of nucleus devices can include static and dynamic axial compression, axial torsion, shear tests, functional range of motion, subsidence, mechanical behavior change due to aging, swelling pressure, and viscoelastic testing. Nucleus devices shall be tested statically to failure and tested cyclically to estimate the maximum run out load or moment at 10 × 10^6^ cycles. Specifies dimensions and materials to create surrogate annuli.
	ASTM F2346-18: Standard Test Methods for Static and Dynamic Characterization of Spinal Artificial Discs [106]	Specifies the materials and methods for the static and dynamic testing of artificial IVDs. Physiological compressive preloads of 500 N lumbar artificial IVDs are required for the static torsion test. Cycle limit of 10 × 10^6^ cycles.
	ASTM F2423-11(2016): Standard Guide for Functional, Kinematic, and Wear Assessment of Total Disc Prostheses [107]	Provides guidance for wear and/or fatigue testing of total IVD prostheses under functional and kinematic conditions and describes test methods for assessment of the wear or functional characteristics, or both, of total IVD prostheses. Flexion/extension—axial load: 1200 N; cyclic axial load (min–max): 900–1850 N; ROM: ±7.5°; moment: ±10 Nm. Rotation—axial load: 1200 N; cyclic axial load (min–max): 900–1850 N; ROM: ±3°; moment: ±10 Nm. Lateral bending—axial load: 1200 N; cyclic axial load (min–max): 900–1850 N; ROM: ±6°; moment: ±12 Nm.
Mechanical (continued)	ASTM F3295-18: Standard Guide for Impingement Testing of Total Disc Prostheses [108]	Provides guidance on the evaluation of wear and fatigue characteristics of total IVD prostheses under cyclic impingement conditions. Axial force: 1200 N; minimum impingement: initial impingement angle, less 2.0°; maximum impingement: ultimate angle plus 2.0°; axial rotation: ±2°.
Biological	BS EN ISO 10993: Biological evaluation of medical devices, Parts 1–20 [109]	Comprehensive guidelines for testing of medical devices comprising the evaluation and testing within a risk management process; animal welfare requirements; test for genotoxicity, carcinogenicity and reproductive toxicity; tests for interaction with blood; tests for in vitro cytotoxicity; tests for local effects after implantation; ethylene oxide sterilization residuals; identification and quantification of potential degradation products; tests for irritation and skin sensitization; tests for systemic toxicity; sample preparation and reference materials; identification and quantification of degradation products from polymeric, ceramic, and metals and alloys; toxicokinetic study design for degradation products and leachables; establishment of allowable limits for leachable substance; chemical characterization of materials; physicochemical, morphological, and topographical characterization of materials; and the principles and methods for immunotoxicology testing.
	ASTM F981-04(2016): Standard Practice for Assessment of Compatibility of Biomaterials for Surgical Implants with Respect to Effect of Materials on Muscle and Insertion into Bone [110]	Provides a series of experimental protocols for biological assays of tissue reaction to nonabsorbable biomaterials for surgical implants. It assesses the effects of the material on animal tissue in which it is implanted. Applies only to materials with projected applications in humans where the materials will reside in bone or soft tissue in excess of 30 days and will remain unabsorbed. Implant shall be made in a cylindrical shape with hemispherical ends and may range from 1 to 6 mm in diameter and from 10 to 20 mm in length depending upon the relative size of the species under study.
Biological (continued)	ASTM F1983-14: Standard Practice for Assessment of Selected Tissue Effects of Absorbable Biomaterials for Implant Applications [111]	Provides experimental protocols for biological assays of tissue reactions to absorbable biomaterials for implant applications. This practice applies only to absorbable materials with projected clinical applications in which the materials will reside in bone or soft tissue longer than 30 days and less than three years.
Materials	ASTM F3142-16: Standard Guide for Evaluation of in vitro Release of Biomolecules from Biomaterials Scaffolds for Tissue-Engineered Medical Products (TEMPs) [112]	Describes general principles of developing and/or using an in vitro assay to evaluate biomolecule release from biomaterials scaffolds for TEMPs that do not contain cells.
	ASTM F2150-19: Standard Guide for Characterization and Testing of Biomaterial Scaffolds Used in Regenerative Medicine and Tissue-Engineered Medical Products [113]	Describes available test methods for the characterization of the compositional and structural aspects of biomaterial scaffolds used in the development and manufacture of regenerative medicine and tissue-engineered medical products (TEMPs).
Imaging	ASTM F2052-15: Standard Test Method for Measurement of Magnetically Induced Displacement Force on Medical Devices in the Magnetic Resonance Environment [94]	Describes the measurement of the magnetically induced displacement force produced by static magnetic field gradients on medical devices and the comparison of that force to the weight of the medical device. It is intended for devices that can be suspended from a string.
	ASTM F3224-17: Standard Test Method for Evaluating Growth of Engineered Cartilage Tissue using Magnetic Resonance Imaging [114]	This standard has been prepared for evaluation of engineered cartilage tissue growth at the preclinical stage and summarizes results from tissue growth evaluation of tissue-engineered cartilage in a few notable cases using water spin–spin relaxation time, T2, in vitro and in vivo in small animal models. Intended to be applicable to most porous natural and synthetic polymers used as a scaffold in engineered cartilage, such as alginate, agarose, collagen, chitosan, and poly-lactic-co-glycolic acid (PLGA).

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
