# Peer review of "Lumbar Intervertebral Disc Herniation: Annular Closure Devices and Key Design Requirements"

_bioengineering, 2022, doi:10.3390/bioengineering9020047_

Round 1

Reviewer 1 Report

well written paper. Nice review of the devices. It was eductaional for me too. 

Author Response

Response to Reviewer 1 Comments
Point 1: Well written paper. Nice review of the devices. It was educational for me too.
Response 1: Thank you for taking the time to review our manuscript. We are pleased that you felt it was a well written paper and provided a nice review of the devices available for annular closure. Your comments are much appreciated. Thank you again for your help.

Reviewer 2 Report

Low back pain (LBP) is a leading cause of disability across the globe. Degenerative intervertebral disc (IVD) disease is a spinal disorder known to cause LBP. On the other hand, IVD herniation leads to degeneration with its prevalence is estimated at 3–5%. Herniated lumbar IVDs and resultant radiculopathy have resulted in almost 15 million office-based physicians visits per year, creating a financial burden on society exceeding US$50 billion in the US annually. Approximately 300,000 lumbar discectomies are performed each year in the USA, making it the most common procedure performed by spine and neurosurgeons. While IVD fusion is the current gold standard to treat degenerated IVDs, annulus closure after discectomy still is a challenge. Whether barricade is successful or not to tackle the problem, its mechanical properties are significantly compared to the native counterpart and can not regenerate IVD. In this sense, the current review paper addresses one of the major concerns and current challenges in LBP treatment. The paper was written well and very well structured. A few minor comments to further enhance the current research are:

  • Please consistently use IVD rather than disc throughout the manuscript.
  • I believe the current paper communicates wider and attracts more readers with different backgrounds if the structure of the IVD was introduced in the background.
  • Please pay attention to the error in line 81. Same as lines 175, 200, 219, and more!
  • I think one missing consideration is the AF structural defects during progression to herniation and whether the AF lamella or the inter-lamellar matrix to be considered? One good paper to read is https://www.mdpi.com/1422-0067/21/14/4889/htm
  • Another direction can be injectable hydrogels; however, their integration with the native tissue is the current drawback.

Reviewer 3 Report

This is an interesting narrative review on current strategies for annular repairs following lumbar discectomy. The article is interesting and well structured and addresses all the main aspects regarding this interesting and highly needed devices.

Some comments:

  • Language editing is advised as minor grammar mistakes are present throughout the text.
  • Reconsider the use of bold and italic styles in the text.
  • Authors should use MDPI style for Table text and captions.
  • Lines 81, 175, 200, 219, 223, 318: please solve the "reference not found" error and replace it with the correct reference.
  • Line 113: please provide the extended form of PET and put it in brackets.
  • Lines 191-193: how can reduced intradiscal pressure decrease disc height, stresses and shear force?

Author Response

Response to Reviewer 3 Comments

Point 1: This is an interesting narrative review on current strategies for annular repairs following lumbar discectomy. The article is interesting and well structured and addresses all the main aspects regarding this interesting and highly needed devices.

Response 1: Thank you for taking the time to read our manuscript and for all your helpful comments. We were pleased to read that you found the manuscript was an interesting narrative review that was well structured and addressed all the main aspects. We have reviewed your comments carefully and have formulated responses and changes to the manuscript. Thank you again for your thorough review of our work – your comments have helped to improve the quality of our paper – we are very grateful for the time that you have spent on this.

Point 2: Language editing is advised as minor grammar mistakes are present throughout the text.

Response 2: Thank you for highlighting that there are some minor English errors in the manuscript. We have carefully checked the text and made revisions to the minor grammar mistakes throughout the text. We hope that the manuscript will now meet the English requirement. We also thank the reviewer for pointing out that the language needed correction.

Point 3: Reconsider the use of bold and italic styles in the text.

Response 3: We have considered your comment on the use of bold and italic. We have gone through the text and removed the bold and italic styles in the text which was summarising key points for the design specification. Some of these points have now been moved to Appendix A (but not put in bold). Thank you for highlighting that this may be unhelpful to the reader.

Point 4: Authors should use MDPI style for Table text and captions.

Response 4: Thank you for highlighting our format error in the manuscript. We have changed the tables within the manuscript to the style indicated in the MDPI template.

Point 5: Lines 81, 175, 200, 219, 223, 318: please solve the "reference not found" error and replace it with the correct reference.

Response 5: Again we thank the reviewer for highlighting these errors.
Unfortunately this seems to be an error with the referencing which occurred in the conversion to pdf format of the manuscript created by MDPI’s system. This error was not present in the word version of the manuscript submitted. We have manually changed the references (not used endnote) to try to eliminate this issue. The authors will also discuss this issue with the journal editorial team to ensure these errors in the MDPI generated PDF are corrected.

Point 6: Line 113: please provide the extended form of PET and put it in brackets.

Response 6: Thank you for highlighting that this abbreviation may be difficult for readers to understand. This was previously defined in section 1.3.1 but we appreciate that readers might skip to sections on specific devices so we have redefined PET as polyethylene terephthalate on line133.

Point 7: Lines 191-193: how can reduced intradiscal pressure decrease disc height, stresses and shear force?

Response 7: Thank you for your insightful question and highlighting that this may not be well explained in the manuscript. We intended to communicate that if IVD pressure is reduced then volume is also reduced leading to a dimensional (height change) in the IVD. Stresses will also increase on the nucleus if pressure is reduced – similar to a car tire, when inflated most of the stress is well distributed through the rubber of the tyre and evenly spread through the whole of the metal wheel part of the wheel, however, if it is deflated then increased normal and shear stress concentrations will be experienced on the metal part of the wheel causing damage. A similar situation occurs in the IVD.
We appreciate that this is not made clear in the manuscript so we have made the following changes (lines 215-218):
“Any reduction in the pressure reduces IVD volume through IVD height reduction; this changes the stress distribution in the IVD causing increases in stress concentrations within the IVD; and leads to increased shear forces in the nucleus [70]”
We thank the reviewer again for their valuable comments which have improved our manscript. We are very grateful for your review